

# Development of a novel embryonic germline gene-related prognostic model of lung adenocarcinoma

Linjun Liu[1], Ke Xu[2] and Yubai Zhou[1]

[1] Department of Biotechnology, College of Life Science & Chemistry, Beijing University of Technology, Chaoyang, Beijing, China
[2] NHC Key Laboratory of Biosafety, China CDC, National Institute for Viral Disease Control and Prevention, Beijing, China

## ABSTRACT

**Background**. Emerging evidence implicates the correlation of embryonic germline genes with the tumor progress and patient's outcome. However, the prognostic value of these genes in lung adenocarcinoma (LUAD) has not been fully studied. Here we systematically evaluated this issue, and constructed a novel signature and a nomogram associated with embryonic germline genes for predicting the outcomes of lung adenocarcinoma.

**Methods**. The LUAD cohorts retrieved from The Cancer Genome Atlas (TCGA) and Gene Expression Omnibus (GEO) database were used as training set and testing set, respectively. The embryonic germline genes were downloaded from the website https://venn.lodder.dev. Then, the differentially expressed embryonic germline genes (DEGGs) between the tumor and normal samples were identified by limma package. The functional enrichment and pathway analyses were also performed by clusterProfiler package. The prognostic model was constructed by the least absolute shrinkage and selection operator (LASSO)-Cox regression method. Survival and Receiver Operating Characteristic (ROC) analyses were performed to validate the model using training set and four testing GEO datasets. Finally, a prognostic nomogram based on the signature genes was constructed using multivariate regression method.

**Results**. Among the identified 269 DEGGs, 249 were up-regulated and 20 were down-regulated. GO and KEGG analyses revealed that these DEGGs were mainly enriched in the process of cell proliferation and DNA damage repair. Then, 103 DEGGs with prognostic value were identified by univariate Cox regression and further filtered by LASSO method. The resulting sixteen DEGGs were included in step multivariate Cox regression and an eleven embryonic germline gene related signature (EGRS) was constructed. The model could robustly stratify the LUAD patients into high-risk and low-risk groups in both training and testing sets, and low-risk patients had much better outcomes. The multi-ROC analysis also showed that the EGRS model had the best predictive efficacy compared with other common clinicopathological factors. The EGRS model also showed robust predictive ability in four independent external datasets, and the area under curve (AUC) was 0.726 (GSE30219), 0.764 (GSE50081), 0.657 (GSE37745) and 0.668 (GSE72094). More importantly, the expression level of some genes in EGRS has a significant correlation with the progression of LUAD clinicopathology, suggesting these genes might play an important role in the progression

Corresponding authors
Ke Xu, xuke@ivdc.chinacdc.cn
Yubai Zhou, zhouyubai@bjut.edu.cn

of LUAD. Finally, based on EGRS genes, we built and calibrated a nomogram for conveniently evaluating patients' outcomes.

## INTRODUCTION

As the most common pathological subtype of non-small cell lung cancer (NSCLC), lung adenocarcinoma (LUAD) largely occurs among women and non-smoking populations (*Herbst, Morgensztern & Boshoff, 2018*). Benefiting from the advance in surgical technique, targeted therapy, immunotherapy and radiochemotherapy, the treatment of LUAD has undergone revolutionary changes in recent years (*Yang, Yang & Yang, 2020*). However, due to the lack of sensitive early detection methods, most patients are in advanced stages at the time of their first diagnosis. The overall 5-year survival rate has not improved significantly (*Chen et al., 2014*).

Many cancers activate genes normally associated with different developmental states, including germ cell-specific genes, which suggests that some unknown mechanisms which are normally limited to the development of germ cells maybe involved in the process of tumorigenesis (*Bruggeman et al., 2018*; *Erenpreisa & Cragg, 2010*; *Kho et al., 2004*; *Lepourcelet et al., 2005*; *McFarlane & Wakeman, 2017*; *Simpson et al., 2005*). With the first discovery of MAGE-1 in melanoma cells, a series of genes that are normally restricted to the testis, such as BAGE, GAGE, HOM-MEL-40, SYCP1, and NY-ESO-1 have been found in various tumors, and these genes are collectively called cancer/testis (CT) genes (*Almeida et al., 2009*; *Boël et al., 1995*; *Chen et al., 1997*; *De Backer et al., 1999*; *Türeci et al., 1996*; *Türeci et al., 1998*; *van der Bruggen et al., 1991*). Although the driving role of these CT genes in tumorigenesis and development has not been fully studied, there is evidence that they may play a role in the early stage of oncogenesis and the maintenance of tumor phenotypes (*Bruggeman et al., 2018*). However, these previously identified CT genes are mixed with genes derived from testicular somatic cells and are not truly germline- specific genes. Currently, using bioinformatic methods, *Bruggeman et al. (2020)* identified 672 so called embryonic germ cell genes which are restricted to the germline and also expressed in a variety of tumor tissues, including lung cancer. However, the level of expression of these genes in lung adenocarcinoma and its relationship with patient prognostics have not been fully studied.

In this study, we systematically analyzed the expression levels of embryonic germline cell-specific genes in LUAD cohort and constructed a novel prognostic signature and nomogram, which can be used to predict the overall survival of LUAD patients.

## MATERIAL AND METHODS

### Data source and preprocessing

Data were collected and preprocessed as previously described (*Yue, Ma & Zhou, 2019*). Briefly, as the training dataset, the expression profile and clinical information of LUAD cohort were downloaded from TCGA database using GDC tool and summarized into an expression and a clinical matrix, respectively. The ensemble ids in the expression matrix were converted into gene symbols according to the annotation file and the multiple expression data of a single gene were replaced by an average value. Genes that are not expressed in all samples are also removed from the expression matrix. For testing sets, we used lung adenocarcinoma as a keyword to search in the GEO database. After removing the datasets which are devoid of clinical data or expression data of genes in the model, four independent GEO datasets GSE30219, GSE50081, GSE37745 and GSE72094 were selected and downloaded via GEOquery package (*Davis & Meltzer, 2007*). For pan-cancer analyses, the expression profiles and clinical data of 33 TCGA cancer types were downloaded from TCGA database, and preprocessed according to previously mentioned procedures. Tumor cohorts with normal samples <5 were not included in the pan-cancer expression analysis. The gene expression values were all converted to transcripts per million transcripts (TPM) to facilitate subsequent analysis. The protein expression data of LUAD were obtained from the Clinical Proteomic Tumor Analysis Consortium (CPTAC) database (https://cptac-data-portal.georgetown.edu/).

### Identification of the differentially expressed embryonic germline genes (DEGGs)

The embryonic germline genes were downloaded from the website https://venn.lodder.dev (*Bruggeman et al., 2020*), and their expression data were extracted into a matrix. Then, the differentially expressed embryonic germline genes (DEGGs) between tumor and normal samples were identified using methods mentioned previously (*Yue, Ma & Zhou, 2019*). The multiple comparisons were controlled using the false discovery rate (FDR) (*Benjamini & Hochberg, 1995*). The screening criteria were |log2Fold Chang| > 1 and FDR < 0.05.

### Functional enrichment and pathway analysis

The Gene Ontology(GO) enrichment and Kyoto Encyclopedia of Genes and Genomes(KEGG) pathway analysis were performed according to previously described methods (*Yu et al., 2012*; *Yue, Ma & Zhou, 2019*). $P$. adjust (FDR) <0.05 was considered statistically significant.

### Construction of EGRS in training dataset

The EGRS model was established using methods described previously (*Yue, Ma & Zhou, 2019*). Briefly, the 454 LUAD samples were included in the univariate Cox regression analysis to determine the prognostic value of DEGGs. $P < 0.05$ was considered statistically significant. The prognostic DEGGs were further screened out using LASSO regression via glmnet package in R software for avoiding model overfitting (*Friedman, Hastie & Tibshirani, 2010*). Then, the resulting key DEGGs were included in multivariate Cox

analysis, and risk score formula was constructed as follow:

$$\text{Risk score} = \sum \beta * \text{gene}.$$

The $\beta$ and gene represent the coefficient of multivariate Cox regression and the expression value of corresponding DEGGs in EGRS, respectively. The $*$ represents the multiplication operator.

## Validation of the EGRS in the training and testing sets

To further validate the expression pattern, the protein expression profiles of DEGGs in EGRS between normal and tumor samples were analyzed using proteomic data of LUAD cohort obtained from CPTAC database. Then, the predicted effect of the EGRS was evaluated in the training and testing sets, respectively. Samples in the training set were divided into high- and low-risk group according to the median of risk score values. The risk score distribution, survival status and the expression level of 11 DEGGs in EGRS between the high- and low-risk samples were evaluated. In addition, we divided the patients in training set into two groups according to commonly used clinicopathological factors and analyze the differentiation of the expression levels of 11 DEGGs between the two groups by student's $t$-test. $P < 0.05$ was considered statistically significant. Kaplan–Meier (KM) survival analysis and receiver operating characteristic (ROC) curve were used to evaluate the predicting power of the EGRS, and the prognostic performance of other clinicopathological factors was also compared with that of EGRS model. The KM plot and ROC curve were used to assess the generalization power of EGRS model in four independent GEO datasets. Finally, a pan-cancer expression and prognostic analyses of 11 DEGGs were performed to extend our findings to other types of cancers.

## Construction and evaluation of a nomogram

For predicting the OS of individual LUAD patient, Nomogram was generated based on the results of the multivariate Cox analysis to predict 1-, 3-, and 5-year OS. The performance of the nomogram was evaluated using calibration plots.

# RESULT

## Identification the DEGGs in LUAD

The transcriptome data of LUAD cohort which contains 497 tumor and 54 normal samples were downloaded from TCGA database, and the corresponding expression profiles of embryonic germline genes were extracted according to the data published previously (*Bruggeman et al., 2020*). The DEGGs between normal and tumor samples were identified by limma package in R software (*Ritchie et al., 2015*). Totally, 269 DEGGs were screened out, among which, the upregulated DEGGs were 249, and the down-regulated DEGGs were 20 (Fig. 1A, Table S1). The representative DEGGs are shown by heat map (Fig. 1B) and box plot (Fig. 1C), respectively.

## Functional enrichment and KEGG analysis of DEGGs

The GO enrichment analysis of the DEGGs revealed that the DEGGs were mainly enriched in biological process of cell proliferation and DNA damage repair, such as nuclear division
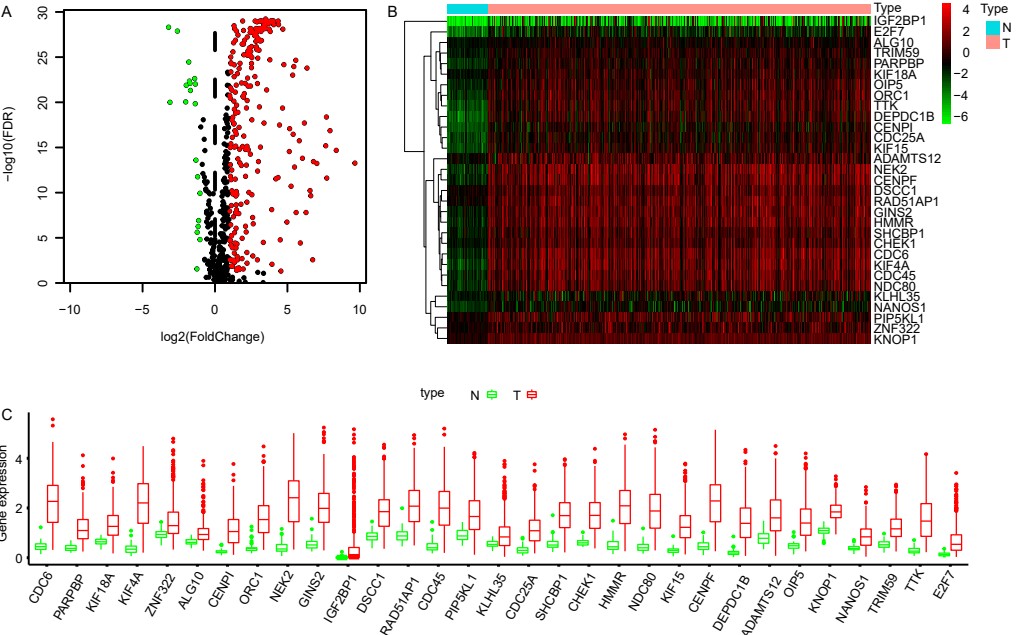

**Figure 1** **The differentially expressed embryonic germline genes (DEGGs) between LUAD and normal tissues.** (A) The volcano plot of DEGGs. The red and green dots represent the up-regulated and down-regulated DEGGs, and the black dots represent the embryonic germline genes that do not meet the selection criteria. (B) The heatmap of the representative DEGGs. N and T represent the normal and LUAD samples, respectively. The color of scale bar (from green to red) indicates the expression status of DEGGs (from lowly expression to highly expression). (C) The boxplot of the representative DEGGs. $P < 0.05$ was considered as statistical significance. LUAD, Lung adenocarcinoma; FDR, False Discovery Rate.

(GO:0000280), DNA replication (GO:0006260), cell cycle checkpoint (GO:0000075), double-strand break repair (GO:0006302) and interstrand cross-link repair (GO:0036297) (Figs. 2A, 2B). The KEGG analysis showed that the enriched pathways were cell cycle (hsa04110), homologous recombination (hsa03440), and Fanconi anemia pathway (hsa03460) (Figs. 2C, 2D). The detailed GO terms of the three categories, biological process (BP), cellular component (CC), molecular function (MF) and KEGG results were presented in Tables S2 and S3, respectively.

## The construction of embryonic germline gene related prognostic model

For EGRS construction, the univariate Cox regression analysis was conducted and the resulting 103 DEGGs with prognostic value ($p < 0.05$, Fig. 3A) were further screened by LASSO regression. Then, the sixteen key DEGGs were selected to performed the multivariate Cox regression analysis (Figs. 3B, 3C). Finally, a prognostic signature containing eleven DEGGs (RAD54L, ZNF322, CENPI, IGF2BP1, IGF2BP3, RAD51AP1, E2F7, HMMR, DNAJC5B, ADAMTS12, NANOS1) was constructed and the risk score formula was presented as follow:

$$\text{Risk score} = (-0.4294) * \text{RAD54L} - 0.2411 * \text{ZNF322} - 0.4645 * \text{CENPI} + 0.2243 *$$

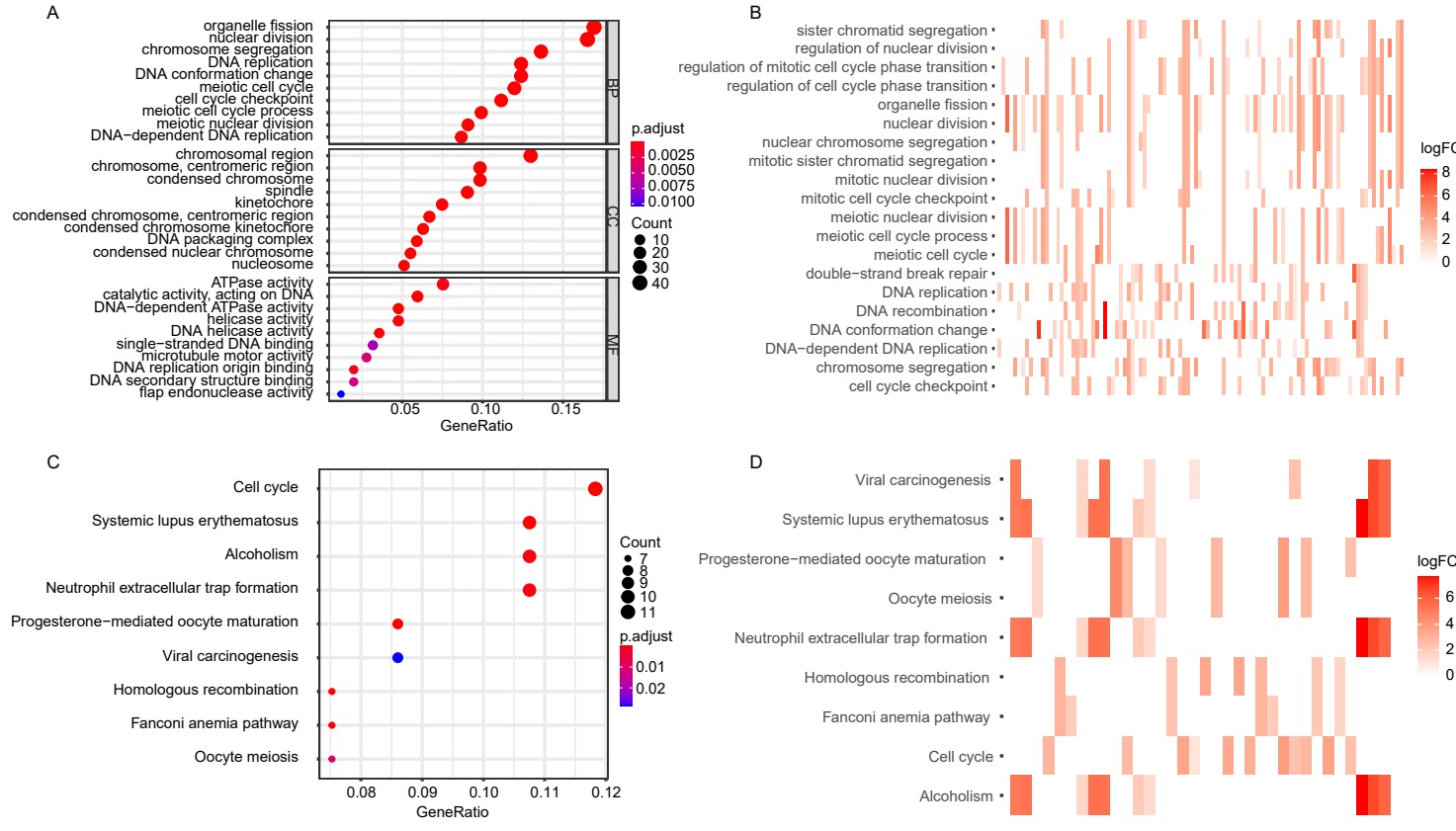

**Figure 2** **The GO and KEGG enrichment analysis of DEGGs.** (A, C) The bubble plot of GO and KEGG pathway analysis of DEGGs, respectively. The color scale represents large (blue) to small (red) p.adjust values. The size of the black dots represents the number of DEGGs enriched in the corresponding terms or pathways. (B, D) The heat plot of GO and KEGG pathway analysis of DEGGs , respectively. The color depth represents the expression status of DEGGs, and the length of the bar represents the number of DEGGs enriched in the corresponding terms or pathways. BP, Biological Process; CC, Cellular Component and MF, Molecular Function; GO, Gene Ontology; KEGG, Kyoto Encyclopedia of Genes and Genomes; logFC, log2FoldChange; DEGGs, differentially expressed embryonic germline genes.

$$IGF2BP1 + 0.2095 * IGF2BP3 - 0.5093 * RAD51AP1 + 0.7979 * E2F7 + 0.7968 *$$
$$HMMR - 0.3272 * DNAJC5B + 0.1997 * ADAMTS12 - 0.5248 * NANOS1.$$

## The validation of the EGRS for survival prediction

Due to the potential inconsistencies in gene expression between the transcriptome and the proteome, we first verified the protein expression of the 11 DEGGs in EGRS in normal and tumor samples through the CPTAC database. Given the limitations of available data, only 4 genes of the 11 DEGGs were analyzed. Consistent with the transcriptome data, the protein expression of 4 genes (IGF2BP1, IGF2BP3, HMMR and ADAMTS12) were up-regulated in LUAD tumor samples. The log2FoldChanges of IGF2BP1, IGF2BP3, HMMR and ADAMTS12 were 0.49, 1.14, 0.50 and 0.87 and the $p$-values of the 4 genes were 2.7E−02, 9.22E−08, 1.32E−07 and 7.27E−18, respectively. In addition, the pan-cancer expression analysis also revealed that 11 DEGGs in EGRS showed differential expression in most types
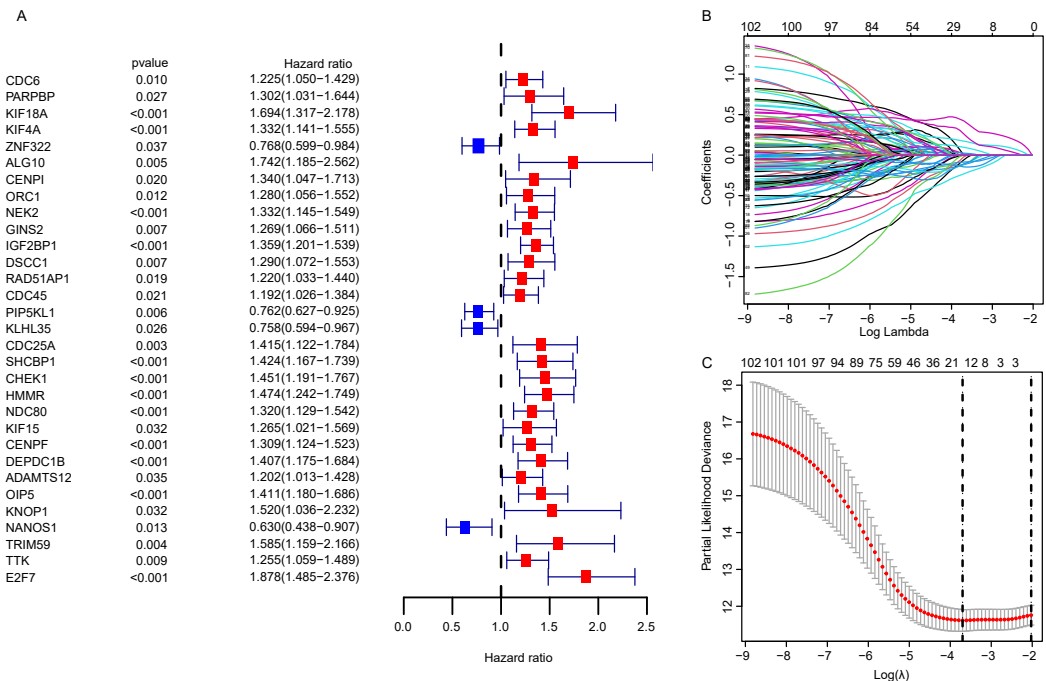

**Figure 3 The identification of the DEGGs with prognostic value.** (A) The forest plot of the representative prognostic DEGGs screened out by univariate Cox proportional hazards regression. The red rectangles represent hazard factors, and blue rectangles represent protective factors. (B) Lasso co-efficient profiles of prognostic DEGGs by optimal lambda. (C) The partial likelihood deviance plot presented the minimum number corresponds to the covariates used for multivariate Cox analysis. DEGGs, differentially expressed embryonic germline genes; Lasso, least absolute shrinkage and selection operator.

of tumors, but diverse expression trends between tumor and paired adjacent normal tissues of some types of cancers were observed, suggesting that the DEGGs may play different roles in the development of different types of tumors (Fig. S1). Then, Kaplan–Meier (KM) survival analysis indicated that the overall survival between the high- and low-risk groups in training set was significantly different, and the low-risk patients had significantly better outcomes than that of high-risk patients ($P = 4.083e-09$) (Fig. 4A). Besides, our model had the best predictive power, compared with other clinicopathological factors such as age, gender, TNM and stage. Its area under curve (AUC) of ROC was 0.790 (Fig. 4B). The DEGGs in EGRS indicated differential expression between high- and low-risk samples (Fig. 4C), and with the increase of risk score, the proportion of death cases was also increasing (Figs. 4D, 4E). Between different pathological groups, the expression levels of genes in EGRS also showed significant difference (Fig. 5). Contrary to CENPI, the expressions of DNAJC5B in elderly patients (>65 years old) were higher than those in younger patients (≤65 years old) (Figs. 5A, 5B). The HMMR, RAD51AP1 were highly expressed in men than in women (Figs. 5D, 5E). ZNF322 was related to T stage, and its expressions in T3-4 were significantly lower than those in T1-2 (Fig. 5G). The CENPI, DNAJC5B, HMMR, RAD51AP1 and ZNF322 were correlated to N stage. Except for ZNF322, other genes were highly expressed in N1-2 stage (Figs. 5I–5L). The expressions of ADAMTS12 in M1 stage

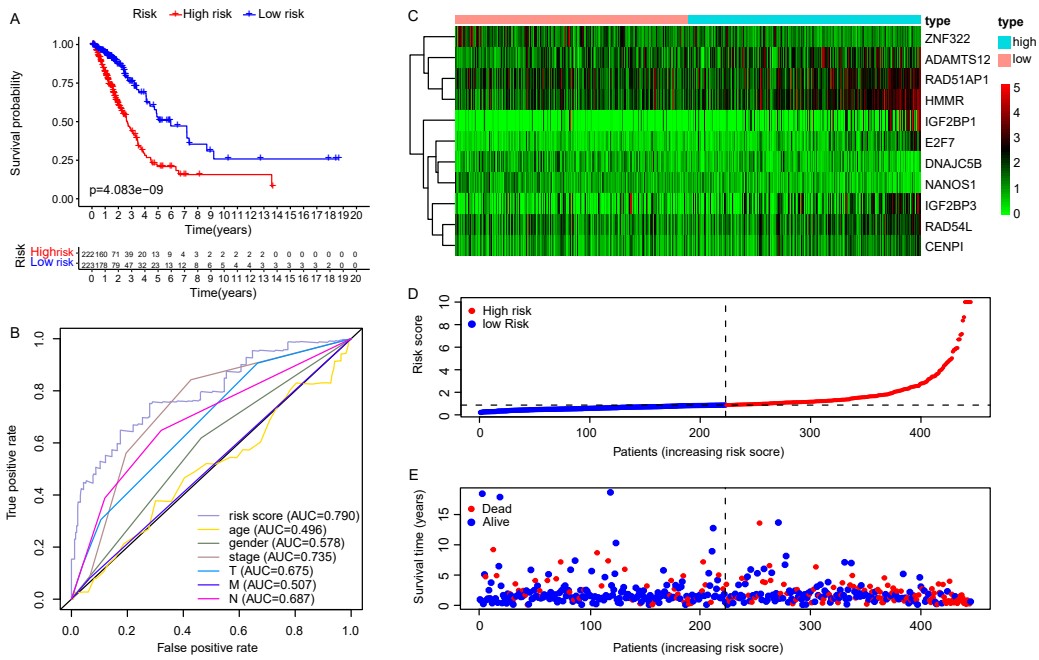

**Figure 4  The construction of embryonic germline gene related prognostic model.** (A) Survival curve of low- and high-risk groups of training set stratified by EGRS. (B) The multi-ROC curve of EGRS and other clinicopathological factors for survival prediction in training set. *P* < 0.05 was considered as statistical significance. (C) The expression status of 11 risk DEGGs in the high-risk (cyan) and low-risk (light red) groups of LUAD patients in training set. (D) The risk score distribution of low-risk (blue) and high-risk (red) groups of LUAD patients in training set. (E) The scatter plot of survival status of LUAD patients in training set. The red dots represent the dead patients, and the blue dots represent the alive patients. LUAD, Lung adenocarcinoma; DEGGs, differentially expressed embryonic germline genes; EGRS, embryonic germline gene related signature; ROC, Receiver Operating Characteristic; AUC, Area Under Curve.

were lower than those in M0 stage (Fig. 5N). The expressions of DNAJC5B, HMMR, IGF2BP1, IGF2BP3, RAD51AP1, ZNF322 were correlated to the pathological stage. Except for ZNF322 and DNAJC5B, the expressions of other genes in stage III-IV were significantly higher than those in stage I-II (Figs. 5P–5U). More importantly, the risk scores in the advanced LUAD were significantly higher than those in the early stage (Fig. 5H, 5M and 5V), although its distribution had no statistical difference in some pathological statuses such as age, gender and tumor metastasis (Figs. 5C, 5F and 5O).

The univariate and multivariate Cox analyses suggested the risk score was an independent prognostic factor (Figs. 6A, 6B). The pan-cancer prognostic analysis also indicated the 11 DEGGs had prognostic value in some cancer types. Likewise, different prognostic effects based on cancer type were observed in some DEGGs (the detail results were packaged into a Supplementary file). In addition, the predictive power of the EGRS was further validated in four GEO testing datasets. Consistent with the training set, our model also stratified the samples in the testing sets into low-risk and high-risk groups, and the low-risk patients had better outcomes (Figs. 6C–6F). The AUC results also showed that our model had stable

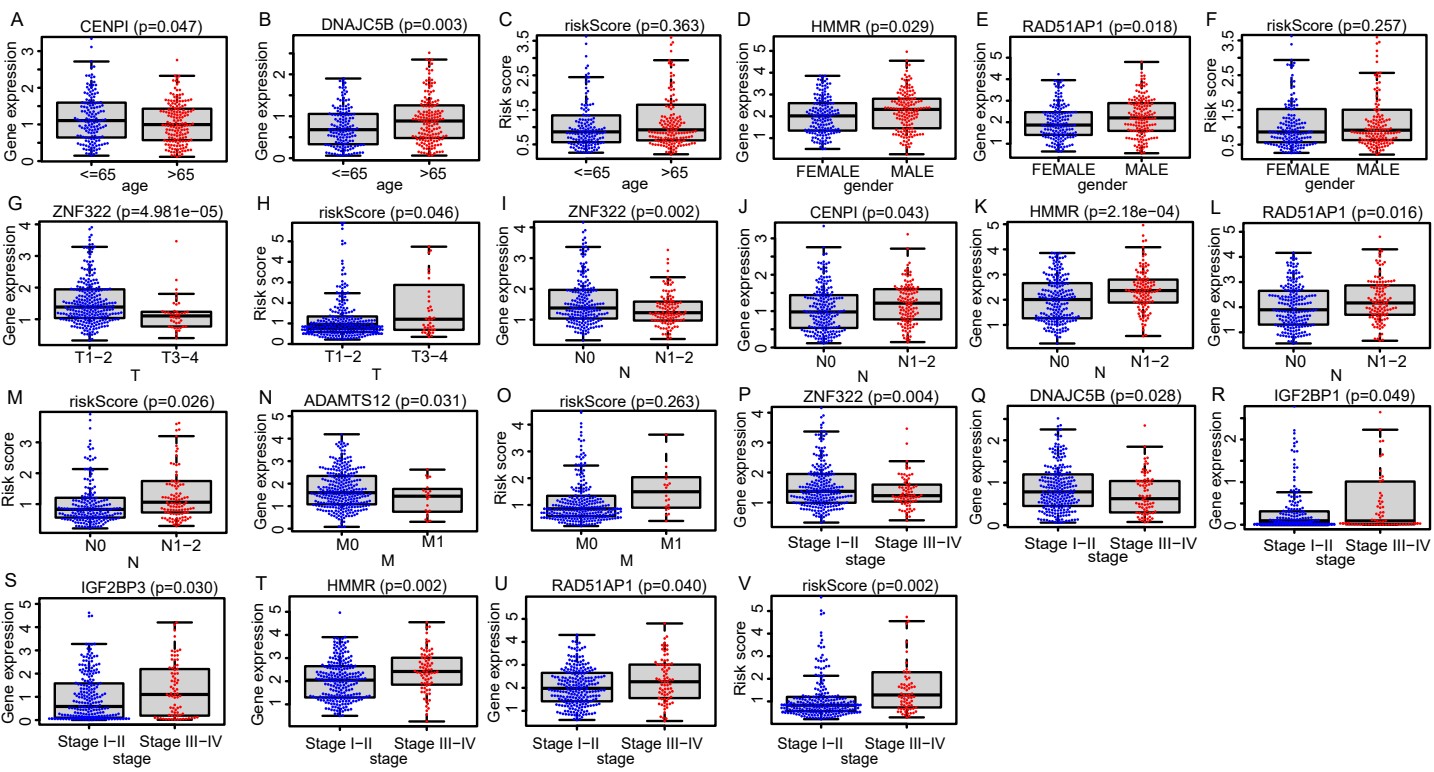

**Figure 5** The distributions of EGRS genes' expressions or risk score to pathological status in training set. (A–C) Age, (D–F) gender, (G, H) T, (I–M) N, (N, O) M and (P–V) stage. EGRS, embryonic germline gene related signature.

predictive ability. The AUC in four independent GEO datasets were 0.726 (GSE30219), 0.764 (GSE50081), 0.657 (GSE37745) and 0.668 (GSE72094) (Figs. 6G–6J).

## Construction and evaluation of a nomogram

In order to more straightforwardly predict the prognosis of LUAD patients, a nomogram was constructed (Fig. 7A). According to the expression level of 11 genes in an individual patient, a numeric value was calculated to directly predict the 1-, 3-year and 5-year prognostic survival rate. The calibration plots showed that the predictive values of OS probabilities in 1-, 3-, and 5-year fit well with the observation values, suggesting that the nomogram is suitable for predicting the prognostic survival rate of LUAD patients (Figs. 7B, 7C, 7D).

## DISCUSSION

Although the data published recently on lung cancer in the United States are encouraging, lung cancer is still a malignant tumor with high morbidity and mortality in the world (*Siegel, Miller & Jemal, 2020*; *Sung et al., 2021*). In China, the incidence of lung cancer ranks the first and the second among men and women, respectively, and it is still the malignant disease with the highest mortality rate (*Zhang et al., 2020b*). The lung adenocarcinoma is the most common pathological subtype of non-small cells lung cancer (NSCLC), which accounts for

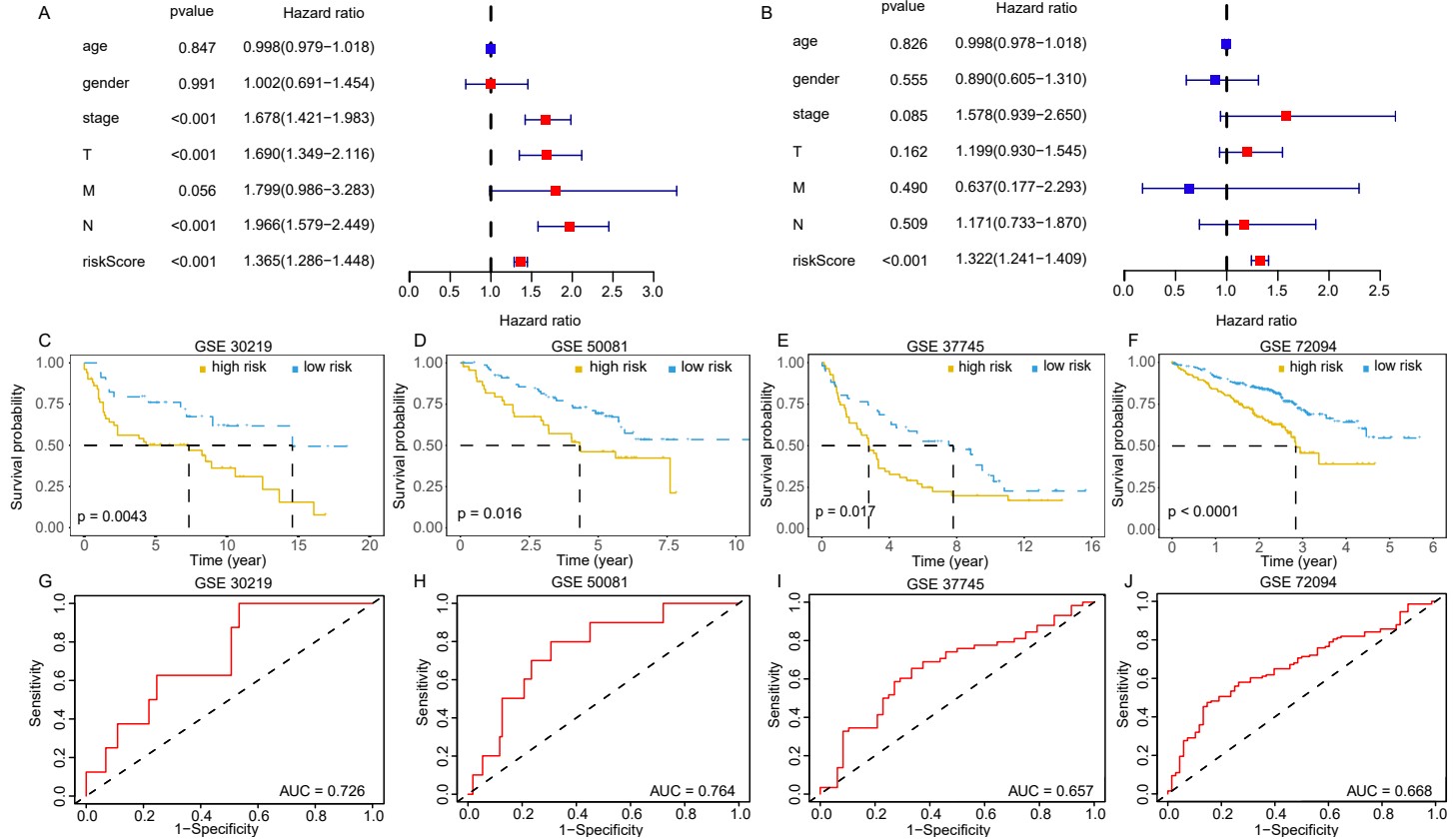

**Figure 6** **The validation of embryonic germline gene related prognostic model.** (A) The prognostic effect analyses of EGRS and commonly used prognostic factors using univariate Cox regression model. (B) The independent prognostic effect analyses of EGRS and commonly used prognostic factors using multivariate Cox regression model. Kaplan-Meier survival curves showing overall survival outcomes of GSE30219 (C), GSE50081 (D), GSE37745 (E) and GSE72094 (F) according to high-risk and low-risk patients stratified by EGRS. The ROC analysis of GSE30219 (G), GSE50081 (H), GSE37745 (I) and GSE72094 (J) for survival prediction by the EGRS. EGRS, embryonic germline gene related signature; ROC, Receiver Operating Characteristic; AUC, Area Under Curve.

about 40% of all lung cancer cases (*Kleczko et al., 2019*). Benefiting from carrying a higher proportion of actionable mutations, lung adenocarcinoma has more treatment options than other subtypes of lung cancer, however, its five-year overall survival rate remains unsatisfactory (*Zhang et al., 2019b*). Accumulating evidence indicates that patients with early stage of disease have a better prognosis, so finding more effective early diagnostic biomarkers or therapeutic targets is important for developing more effective treatment regimens, prolonging patient's survival time and improving their quality of life.

The various types of human cells are derived from a single fertilized egg cell through differentiation mechanisms that we do not fully understand (*McKenna & Gagnon, 2019*). In this process, specific types of genes in differentiating cells are expressed in a well-organized spatiotemporal sequence under the cues of various internal and external factors, and finally generate tissue cells with distinct morphology, function and gene expression profiles (*Trott & Martinez Arias, 2013*). Germline cell-specific genes are a class of genes that are expressed only in germ cells. Accumulating evidence shows that tumor cells and germ cells have

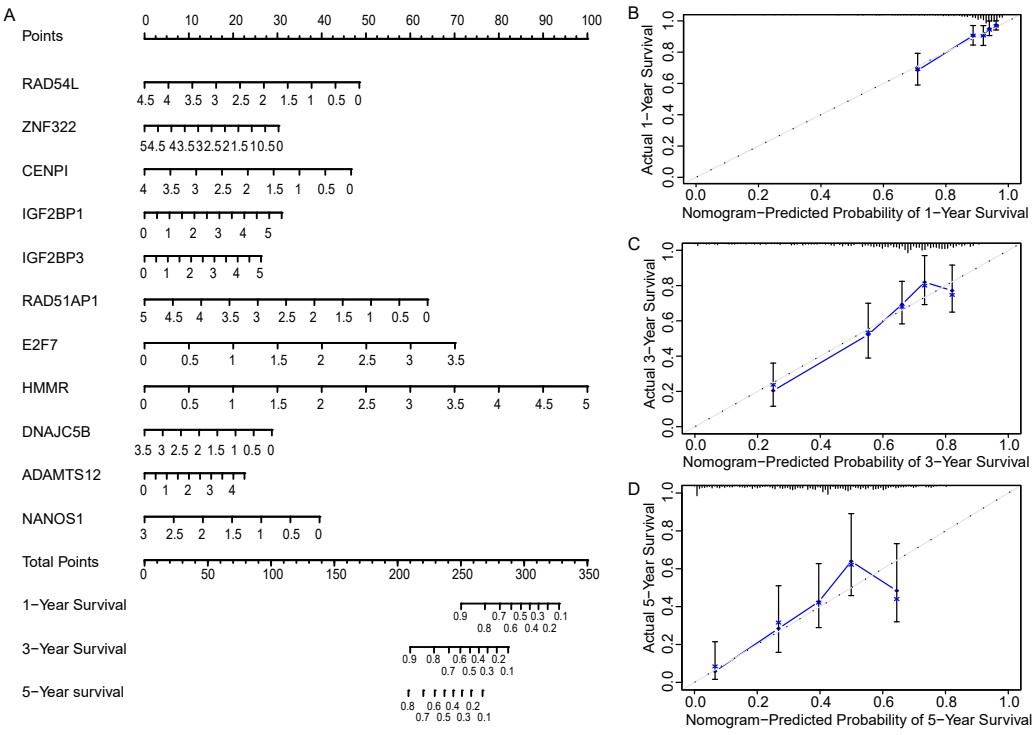

**Figure 7** **Prognostic nomogram and calibration plots of EGRS.** (A) The nomogram of EGRS. (B, C and D) calibration plots of nomogram predicting 1-year, 3-year and 5-year survival, respectively.

similar behaviors, so it is speculated that tumor cells may adopt certain similar regulatory mechanisms in the process of tumorigenesis and development (*Bruggeman et al., 2018*; *Simpson et al., 2005*). Indeed, follow-up studies have detected multiple germ cell-specific genes in different tumor tissues (*Erenpreisa & Cragg, 2010*; *McFarlane & Wakeman, 2017*; *Whitehurst, 2014*). The products of these genes are regarded as ideal immunotherapy targets due to their restricted range of expression and have attracted more and more attention.

Currently, *Bruggeman et al. (2020)* identified 672 true embryonic germ cell-specific genes by removing contaminated somatic genes. However, the expression status and their prognostic value of these genes have not been fully studied in lung adenocarcinoma. To take these issues, we systematically studied the expression levels of the above-mentioned 672 genes in the TCGA LUAD cohort, evaluated their correlation with the prognosis of LUAD patients, and finally constructed and validated a prognostic model consisting of 11 DEGGs. Our model can well stratify the patients into high- and low-risk groups and robustly predict the outcomes of patients in both training and testing sets. Among the 11 DEGGs, ZNF322 (also known as ZNF322A, ZNF388 or ZNF489) is a member of the zinc finger transcription factor family, and may act as a positive regulator in MAPK signaling pathway (*Li et al., 2004*). Recent studies have shown that the overexpression of ZNF322 is related to the oncogenesis of many tumors including lung cancer through a variety of mechanisms, such as directly suppressing the expression of c-Myc to promote

cell stemness or up-regulating the transcription of α-adducin (ADD1) and cyclin D1 (CCND1) to promote tumor growth and metastasis (*Jen et al., 2016*; *Jen et al., 2019*; *Liao et al., 2017*; *Williams et al., 2014*). DNA binding protein RAD51AP1 is a component of DNA damage repair system, which plays an important role in RAD51-mediated homologous recombination (*Pires, Sung & Wiese, 2017*). Further studies indicates that overexpression of RAD51AP1 promotes the proliferation of lung cancer cells and correlates to poor prognosis (*Chudasama et al., 2018*; *Li et al., 2018b*; *Wu et al., 2019*). HMMR coding a 724 amino acids protein, which may associate with the motility of cancer cells (*Sankaran et al., 2012*). Quite a few of studies had shown that the highly expression of HMMR is closely related to the prognosis of LUAD patients, and it may promote the progression of lung adenocarcinoma by regulating the metabolic state of tumor cells (*He & Zuo, 2019*; *Jiao et al., 2020*; *Li et al., 2020a*; *Li, Qi & Li, 2020c*; *Liu et al., 2019*; *Man et al., 2014*; *Shen et al., 2019*; *Stevens et al., 2017*; *Zhang, Zhang & Yu, 2019a*; *Zhou et al., 2015*). The ADAMTS12 is a novel anti-tumor metalloprotease, and the expression is epigenetically silenced in tumor cells (*Li et al., 2018a*; *Moncada-Pazos et al., 2009*). A bulk of studies suggest the overexpression of IGF2BP1 and IGF2BP3 facilitate the progress of lung cancer, and relate to the prognosis of LUAD patients (*Beljan Perak et al., 2012*; *Gong et al., 2016*; *Guo et al., 2021*; *Huang et al., 2019*; *Kato et al., 2007*; *Li & Zhan, 2020*; *Li et al., 2020b*; *Mizutani et al., 2016*; *Müller et al., 2019*; *Ohdaira et al., 2012*; *Wang, He & Ma, 2020*; *Xueqing et al., 2020*; *Zhang et al., 2020a*). E2F7 plays an important physiological role in embryonic development and cell cycle regulation (*Park et al., 2015*). Recent researches show the various non-coding RNA can influence the progress of lung cancer by regulating the expression of E2F7 (*Liang et al., 2018*; *Liu et al., 2020*; *Wang et al., 2021*; *Yuan et al., 2021*). NANOS1 was up-regulated in lung cancer, and promoted the progress of tumor progress (*Bonnomet et al., 2008*; *De Keuckelaere et al., 2018*). RAD54L is also a potential prognostic biomarker of NSCLC (*Tu et al., 2021*; *Zheng et al., 2021*). Literature retrieval showed that the relation of LUAD and DEGGs DNAJC5B, CENPI has not been reported so far. In addition, pan-cancer expression and prognosis analyses also indicated that 11 DEGGs showed differential expression and prognostic value in a variety of tumors, suggesting that these DEGGs may also play a role in the development of tumors other than lung adenocarcinoma, which needed to be validated by further functional experiments.

Although we constructed and validated an 11-gene prognostic model, the present study still has several limitations: first, the number of samples in training set is not large enough; secondly, due to tumor heterogeneity and individual differences among LUAD patients, the generalizing abilities of the model need further improvement. Therefore, further model optimization and validation using LUAD cohort with larger samples are needed. In addition, the results of bioinformatics analysis did not carry out related experimental verifications. Therefore, whether the genes in the model play a functional role in the progress of LUAD remains to be resolved. These issues mentioned above will be the main content of our follow-up studies.

## CONCLUSION

In summary, we used LUAD transcriptome data to identify DEGGs. From the DEGGs, an 11-gene signature and a prognostic nomogram were constructed and validated for predicting the outcomes of LUAD patients. Further studies on these genes will provide a new insight into the potential relationship between tumor microenvironment and LUAD prognosis.

### Funding

This work was supported by the Major National Research and Development Projects (2012ZX10001005-007, 2018ZX10731101-002-007, 2018ZX10731101-001-017); the National High Technology Research Program of China (2012AA02A404); and the State Key Laboratory for Infectious Disease Prevention and Control (2011SKLID103). The funders had no role in study design, data collection and analysis, decision to publish, or preparation of the manuscript.

### Grant Disclosures

The following grant information was disclosed by the authors:
Major National Research and Development Projects: 2012ZX10001005-007, 2018ZX10731101-002-007, 2018ZX10731101-001-017.
National High Technology Research Program of China: 2012AA02A404.
State Key Laboratory for Infectious Disease Prevention and Control: 2011SKLID103.

### Competing Interests

The authors declare there are no competing interests.

### Author Contributions

- Linjun Liu performed the experiments, analyzed the data, prepared figures and/or tables, authored or reviewed drafts of the paper, and approved the final draft.
- Ke Xu conceived and designed the experiments, analyzed the data, prepared figures and/or tables, authored or reviewed drafts of the paper, and approved the final draft.
- Yubai Zhou conceived and designed the experiments, performed the experiments, analyzed the data, prepared figures and/or tables, authored or reviewed drafts of the paper, and approved the final draft.

### Data Availability

The raw data are available at TCGA and GEO.

TCGA: TCGA-LUAD, TCGA-LUSC, TCGA-BRCA, TCGA-GBM, TCGA-OV, TCGA-UCEC, TCGA-KIRC, TCGA-HNSC, TCGA-LGG, TCGA-THCA, TCGA-PRAD, TCGA-SKCM, TCGA-COAD, TCGA-STAD, TCGA-BLCA, TCGA-LIHC, TCGA-CESC, TCGA-KIRP, TCGA-SARC, TCGA-LAML, TCGA-PAAD, TCGA-ESCA, TCGA-PCPG, TCGA-READ, TCGA-TGCT, TCGA-THYM, TCGA-KICH, TCGA-ACC, TCGA-MESO, TCGA-UVM, TCGA-DLBC, TCGA-UCS and TCGA-CHOL.

GEO: GSE30219, GSE50081, GSE37745 and GSE72094.

## Supplemental Information

Supplemental information for this article can be found online at http://dx.doi.org/10.7717/peerj.12257#supplemental-information.

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
