# Peer review of "Development of a novel embryonic germline gene-related prognostic model of lung adenocarcinoma"

_PeerJ, doi:10.7717/peerj.12257_

## Round 0.1 · original submission · Major Revisions

The reviewers have provided several constructive and insightful comments which I expect to be addressed by authors in order to improve their manuscript.

Reviewer 1 ·

Basic reporting

In the submitted manuscript Linjun Liu et al have done a commendable job performing integrated bioinformatic analysis on embryonic germline genes in relation to lung adenocarcinoma (LUAD) development. The authors pose a logical and relevant question at the introduction and designed rigorous experiments to develop a risk score, by putting appropriate weightage on differentially expressed germline genes that can be used as a prognostic biomarker for LUAD development.

Experimental design

The authors performed rigorous experiments to address a relevant query. The methodology is substantially detailed. Although, I would strongly encourage the authors to describe in detail the methodology of developing the risk score. In the method section, it is not fully clear how the risk score was constructed.

Validity of the findings

Though a similar bioinformatic approach is being taken to develop biomarkers for different cancer progression models, the development of risk-score showed novelty from the research group’s point.

Some minor comments to be addressed by the authors –

Figure 1: Legend – ‘FDR’ full form – False Discovery Rate. As much as I understand that the False positive rate conveys a similar meaning, please correct the abbreviation.

Material and Methods: Line 132 - “The performance of the nomogram was evaluate…….” Please change it to “ was evaluated…”

Please consider rearranging figures 4, 5, and 6 to have a better flow with the text.

Please rearrange figure 5 to have similar pathological status distributions in the same row for better readability. Please provide the distributions of risk scores to all the pathological statuses in the training sets to show parallel comparisons to the distribution of representative ERGS genes’ expressions.

Reviewer 2 ·

Basic reporting

The authors outline methodologies to identify genes, specifically embryonic germline related genes, which can be used as prognostic predictors of lung adenocarcinoma. The article is fairly comprehensible with the article sections, figures and raw data in line with the journal standards. There are few minor grammatical corrections which have been marked in the manuscript, which should be addressed by the authors.

Experimental design

The authors have used several lines/methods of investigations in order to arrive at the conclusions.
The study is supported by relevant analysis, explanation of the techniques used and their relevance.

Validity of the findings

Overall, the potential implications of the study would be of interest in the field of LUAD and it prognostic significance. However, there are a few concerns regarding the results, which the authors must look into addressing (detailed in the comments section).

Additional comments

The study results are promising in terms of potential future studies related to the prognostic indicators for lung adenocarcinoma. The authors are suggested to address the following in order improve the overall impact of the study.
Minor suggestions:
The authors are urged to rectify certain grammatical errors and revise few sentences in the following lines of the manuscript (the specific edits are highlighted in the attached manuscript file)
Lines 59, 69, 151, 225, 251, 255 and 279.

Major concerns:
(1) Although, the authors provide a rationale for studying LUAD, it is recommended that the authors show the relevance of the 11 genes they have identified to be of prognostic value, in other tumor types as well. The studies by Bruggeman et al have already identified several germ-line associated genes in various cancer cell lines. The authors should at the very least look into the expression profiles of the 11 genes in the cancer cell line databases to identify the impact and relevance of their findings. Can the findings be applicable to any other tumor types in addition to lung cancer? Since the genes were identified by taking into account the embryonic germline databases, the authors are especially encouraged to state their thoughts and potential impact of their findings in cancers specifically associated with the reproductive organs (ovarian, endometrium, prostrate cancers to name a few)
(2) The authors have constructed a prognostic model related to the embryonic germline genes (Fig. 4). Can the authors comment on the number of relapse-free events in the patient sets (training set)?
(3) Can the authors clarify how many patients were considered for the nomogram depicted in Fig. 7? The authors state that the expression of 11 genes were calculated from an individual patient. If this is indeed the case, the authors must construct nomograms for a statistically significant number of patients in order to make any claims regarding prediction of survival rates of LUAD patients.
(4) The authors must show protein expression profiles for the 11 genes (or atleast few of the genes) they have identified, either through in vitro experiments measuring protein levels in tumor samples or cancer cell lines or through protein databases. There are several studies showing poor correlation between gene expression data and the corresponding protein expression patterns (such as the study by Vogel, C. et al., Nat. Rev. Genet. 2012, 13, 227–232). The relative protein expressions levels will be one way to provide proof of the functional relevance of the expression of the identified genes to the progression of the disease.

Annotated reviews are not available for download in order to protect the identity of reviewers who chose to remain anonymous.

---

## Round 0.2 · accepted · Accept

Dear Authors, your revised manuscript has been positively evaluated by the reviewers and it is now suitable for publication in PeerJ. Congratulations.

Reviewer 1 ·

Basic reporting

In the revised manuscript the authors Linjun Liu et al have established a risk score, considering appropriate weightage on differentially expressed germline genes to use as a potential biomarker for Lung Adenocarcinoma development.

Experimental design

No comments

Validity of the findings

No Comments

Additional comments

The revised manuscript is stronger with the suggested changes.